

# Whole genome sequencing of a novel, dichloromethane-fermenting *Peptococcaceae* from an enrichment culture

Sophie I. Holland[1,*], Richard J. Edwards[2,*], Haluk Ertan[3,4], Yie Kuan Wong[2], Tonia L. Russell[5], Nandan P. Deshpande[2], Michael J. Manefield[1,4] and Matthew Lee[1]

[1] UNSW Water Research Centre, School of Civil and Environmental Engineering, University of New South Wales, Sydney, New South Wales, Australia
[2] School of Biotechnology and Biomolecular Sciences, University of New South Wales, Sydney, New South Wales, Australia
[3] Department of Molecular Biology and Genetics, Istanbul University, Istanbul, Turkey
[4] School of Chemical Engineering, University of New South Wales, Sydney, New South Wales, Australia
[5] Ramaciotti Centre for Genomics, University of New South Wales, Sydney, New South Wales, Australia
[*] These authors contributed equally to this work.

Corresponding author
Matthew Lee, mattlee@unsw.edu.au

## ABSTRACT

Bacteria capable of dechlorinating the toxic environmental contaminant dichloromethane (DCM, $CH_2Cl_2$) are of great interest for potential bioremediation applications. A novel, strictly anaerobic, DCM-fermenting bacterium, "DCMF", was enriched from organochlorine-contaminated groundwater near Botany Bay, Australia. The enrichment culture was maintained in minimal, mineral salt medium amended with dichloromethane as the sole energy source. PacBio whole genome SMRT™ sequencing of DCMF allowed *de novo*, gap-free assembly despite the presence of cohabiting organisms in the culture. Illumina sequencing reads were utilised to correct minor indels. The single, circularised 6.44 Mb chromosome was annotated with the IMG pipeline and contains 5,773 predicted protein-coding genes. Based on 16S rRNA gene and predicted proteome phylogeny, the organism appears to be a novel member of the *Peptococcaceae* family. The DCMF genome is large in comparison to known DCM-fermenting bacteria. It includes an abundance of methyltransferases, which may provide clues to the basis of its DCM metabolism, as well as potential to metabolise additional methylated substrates such as quaternary amines. Full annotation has been provided in a custom genome browser and search tool, in addition to multiple sequence alignments and phylogenetic trees for every predicted protein, http://www.slimsuite.unsw.edu.au/research/dcmf/.

## INTRODUCTION

Dichloromethane (DCM, $CH_2Cl_2$) is a toxic environmental contaminant. Approximately 70% of all DCM worldwide is of anthropogenic origin, due to its extensive use in industry as

a solvent and aerosol propellant (*Gribble, 2009*; *Marshall & Pottenger, 2016*). It is currently present at 30% of Superfund National Priority List sites within the United States and its territories (*US National Library of Medicine, 2019*), and global capacity for DCM continues to steadily increase (*Marshall & Pottenger, 2016*). As well as being harmful to human health (*Agency for Toxic Substances and Disease Registry, 2000*), DCM has recently been recognised as a potent greenhouse gas (*Hossaini et al., 2017*).

DCM in groundwater can be transformed by both aerobic and anaerobic bacteria, although the former has been more comprehensively characterized (*Leisinger & Braus-Stromeyer, 1995*). Aerobic DCM metabolism is found in facultative methylotrophs, which use a DCM dehydrogenase from the glutathione S-transferase family to catalyse dehalogenation (*Leisinger & Braus-Stromeyer, 1995*). Anaerobically, DCM can be transformed under denitrifying conditions (*Melendez, Roman & Smith, 1993*; *Freedman, Smith & Noguera, 1997*), or co-metabolically under methanogenic conditions (*Freedman & Gossett, 1991*; *Stromeyer et al., 1991*). To date, however, only two DCM-fermenting bacteria have been described and sequenced: *Dehalobacterium formicoaceticum* (*Mägli, Wendt & Leisinger, 1996*; *Chen et al., 2017*) and '*Candidatus* Dichloromethanomonas elyunquensis' (*Kleindienst et al., 2016*; *Kleindienst et al., 2017*). Of these, only the former has been isolated (*Mägli, Wendt & Leisinger, 1996*).

Both species are acetogenic and are thought to metabolise DCM via incorporation of the methyl group into the Wood-Ljungdahl pathway (reviewed in *Ragsdale & Pierce, 2008*), although the precise mechanism of dechlorination has thus far eluded description (*Mägli, Wendt & Leisinger, 1996*; *Kleindienst et al., 2017*). The Wood-Ljungdahl pathway is an ancient metabolism that is present in acetogenic bacteria and methanogenic archaea, and links carbon fixation with ATP generation (*Fuchs, 2011*; *Poehlein et al., 2012*). It is thought that an as-yet unidentified methyltransferase (*D. formicoaceticum*) possibly in concert with a reductive dehalogenase ('*Ca.* Dichloromethanomonas elyunquensis') is responsible for mediating the transformation of DCM into methyl- or methylene-tetrahydrofolate (*Kleindienst et al., 2019*).

In order to investigate possible pathways for DCM transformation within the novel organism described in this paper, we determined that a detailed and accurate genome annotation was necessary. As it can be difficult to assemble a high quality genome from a mixed culture, we sought to overcome these challenges with a thorough genome sequencing and assembly strategy. We report the whole genome sequencing and assembly of a novel, DCM-fermenting bacterium, herein referred to as DCMF. The organism exists in an enrichment culture, "DFE" (DCM-fermenting enrichment), derived from a previously reported methanogenic, DCM-dechlorinating consortium, DCMD (*Lee et al., 2012*). DCMD was dominated by a *Dehalobacter* species whose growth was linked to DCM metabolism. The enrichment process reported here led to a shift from this *Dehalobacter* species to DCMF as the dominant member of the community. From a bioremediation perspective, DCMF is an important addition to the limited group of organisms able to utilize the common environmental pollutant DCM.
## MATERIALS & METHODS

### Inoculum origin

The original inoculum was obtained from sediment drilled from five m beneath the surface of an organochlorine-contaminated coastal sand bed aquifer (Botany Sands aquifer), latitude $-33°57'27.6''$S, longitude $151°12'60.0''$E. The initial, methanogenic enrichment culture using DCM as the sole energy source was reported previously (*Lee et al., 2012*).

### Culture media

Cultures were grown in anaerobic minimal mineral salts medium that comprised (g l$^{-1}$): $CaCl_2.2H_2O$ (0.1), KCl (0.1), $MgCl_2.6H_2O$ (0.1), $NaHCO_3$ (2.5), $NH_4Cl$ (1.5), $NaH_2PO_4$ (0.6), 1 ml of trace element solution A (1,000×), 1 ml of trace element solution B (1,000×), 1 ml of vitamin solution (1,000×), 10 ml of 5 g l$^{-1}$ fermented yeast extract (FYE; 100×), and resazurin 0.25 mg l$^{-1}$. Trace element solutions A and B were prepared as described previously (*Wolin, Wolin & Wolfe, 1963*), as was the vitamin solution (*Adrian et al., 1998*). Medium was sparged with $N_2$ during preparation and the pH was adjusted to 6.8–7.0 by a final purge with $N_2/CO_2$ (4:1). Aliquots were dispensed into glass serum bottles that were crimp sealed with Teflon faced rubber septa (13 mm diameter, Wheaton, Millville, NJ, USA) before the medium was chemically reduced with sodium sulphide (0.2 mM). DCM (1 mM) was supplied as the sole electron source via a glass syringe. Methanogenic Archaea present in the early enrichment culture were inhibited with 2-bromoethanosulfonate (BES, 0.2 mM) for two generations. All cultures were incubated statically at 30 °C in the dark.

### Preparation of spent media as a co-factor solution

A stock FYE solution was prepared by inoculating anoxic yeast extract (5 g l$^{-1}$) in defined minimal mineral salts medium (described above, excluding DCM) with the DFE culture. The culture was incubated for one week at 30 °C before being filter-sterilised. The filtered, spent media was re-inoculated with DFE and incubated for a further week, to ensure that growth was no longer possible on FYE (i.e., that it had been energetically exhausted). The spent media was then filter-sterilised again before use.

### Analytical methods

DCM and methane were quantified on a GS-Q column (30 m × 0.32 mm; Agilent Technologies) using a Shimadzu GC-2010 gas chromatograph with flame ionisation detector (GC-FID). Headspace samples (100 µl) were withdrawn directly from culture flasks using a lockable, gas-tight syringe and injected manually. The oven was initially 150 °C, then raised by 30 °C min$^{-1}$ to 250 °C. The inlet temperature was 250 °C, split ratio 1:10, FID temperature 250 °C. A minimum three-point calibration curve was used. DCM concentrations are reported as the nominal concentration in each serum bottle, calculated from the headspace concentration using the Henry's Law dimensionless solubility constant ($H^{cc} = 0.107$ at 30 °C), as per the OSWER method (*US EPA, 2001*).

### Genomic DNA extraction

Genomic DNA was extracted as previously described (*Urakawa, Martens-Habbena & Stahl, 2010*). Briefly, cells were lysed with lysis buffer and bead-beating, before DNA was extracted

with phenol-chloroform-isoamyl, precipitated using isopropanol, and resuspended in molecular grade water. The nucleic acid concentration was quantified using a Qubit instrument and assay as per the manufacturer's instructions (Life Technologies, Carlsbad, CA, USA).

## Community analysis

Throughout the initial transfers and serial dilutions of the enrichment culture, the community was monitored via denaturing gradient gel electrophoresis (DGGE). DNA was amplified with primers GC338F and 530R (Table S1). DGGE was performed with a DCode mutation detection system (Bio-Rad, Hercules, CA, USA) and a Cipher Electrophoresis system (CBS Scientific Company Inc, San Diego, CA, USA) in a $1\times$ TAE buffer at pH 7.5. PCR products were loaded onto a 10% (v/v) acrylamide gel with a 30–60% gradient of urea and deionised formamide before electrophoresis at 60 °C, 75V for 16.5 h. Gels were stained with SYBR Gold (Invitrogen$^{TM}$, Life Technologies, Carlsbad, CA, USA) in $1\times$ TAE buffer for 10 min, prior to visualisation on a Gel Doc XR (Bio-Rad). Bands of interest were excised, DNA eluted from them in molecular grade water and re-amplified using the 338F primer (Table S1). PCR products were cleaned with a Clean and Concentrate-25 kit (Zymo Research, Irvine, CA, USA).

To confirm the absence of an archaeal population following amendment of the enrichment culture with BES, archaeal specific primers Arc340F and Arc1000R (Table S1) were used for PCR on a T100$^{TM}$ thermal cycler (Bio-Rad).

Quantitative PCR of the *Dehalobacter* spp. 16S rRNA gene was carried out on a CFX96 thermal cycler (Bio-Rad, Table S1). Standards ranged from $10^3$–$10^9$ copies ml$^{-1}$ and were created using serial 10-fold dilutions of a plasmid carrying the cloned gene, constructed with TOPO TA Cloning Kit (Life Technologies).

## Illumina genome sequencing

DNA was prepared with the Nextera XT library prep kit (Illumina, San Diego, CA, USA). Sequencing was carried out on an Illumina MiSeq with a v2 500-cycle kit ($2 \times 250$ bp run) at the Ramaciotti Centre for Genomics (UNSW Sydney, Australia). Three MS110-2 libraries were used for the run. Library size ranged from 200–3,000 bp, with an average of 955 bp. Raw reads were trimmed and filtered with SolexaQA (DynamicTrim.pl and LengthSort.pl) (*Cox, Peterson & Biggs, 2010*). Raw reads were submitted to the NCBI Sequence Read Archive with the identifier SRR5179547.

## Pacific Biosciences SMRT sequencing

A MagAttract HMW DNA kit (Qiagen, Hilden, Germany) was used to extract high-molecular weight genomic DNA, followed by purification using AMPure PB beads (Beckman Coulter, Brea, CA, USA). DNA concentration and purity were checked by Qubit and NanoDrop instruments, respectively. A 0.75% Pippin Pulse gel (Sage Science, Beverly, MA, USA) was performed by the Ramaciotti Centre for Genomics (UNSW Sydney, Australia) to further verify integrity. A SMRTbell library was prepared with the PacBio 20 kb template protocol excluding shearing (Pacific BioSciences, Menlo Park, CA, USA).

Additional damage repair was carried out following minimum 4 kb size selection using Sage Science BluePippin.

Whole genome sequencing was performed on the PacBio RS II (Pacific Biosciences), employing P6 C4 chemistry with 240 min movie lengths. DNA was initially sequenced using two Single Molecule Real Time$^{TM}$(SMRT) cells. A third SMRT$^{TM}$ cell was added to compensate for low quality data from the first two, due to degraded DNA yield from the sample. The SMRTbell library for this cell was prepared with the PacBio 10 kb template protocol, without size selection, and a lower input (3,624 ng) of DNA was used. In total, the three SMRT cells yielded 463,878 subreads from 169,180 ZMW, with a combined length of 1,712,588,985 bp. Reads were submitted to the NCBI Sequence Read Archive with the identifier SRR5179548.

## Genome assembly and annotation

PacBio subreads were assembled using HGAP3 (*Chin et al., 2013*) as implemented in SMRT Portal. In-house software, SMRTSCAPE (SMRT Subread Coverage & Assembly Parameter Estimator; http://rest.slimsuite.unsw.edu.au/smrtscape) was used to predict optimal HGAP settings for several different assemblies with different predicted genome size and minimum correction depths (Table S2). The assembly with the greatest depth of coverage used for seed read error correction that still yielded a full-length (6.44 Mb) intact chromosome was selected for the draft genome. This corresponded to: min read length 4,010 bp; min seed read length 8,003 bp; min read quality 0.86; min 10× correction coverage. The genome was corrected with Quiver (*Chin et al., 2013*) using all subreads and circularised by identifying and trimming overlapping ends, then annotated in-house using Prokka (*Seemann, 2014*).

Based on draft annotation, the genome was re-circularised to have its break-point in the intergenic region between the 3′of two hypothetical genes, and then subjected to a second round of Quiver correction to make sure the manually joined region was of high quality. Filtered Illumina reads were mapped onto the Quiver-corrected genome using BWA-MEM v0.7.9a (*Li, 2013*) and possible errors were identified with Pilon (*Walker et al., 2014*). Manual curation was then performed to check any discrepancies between the PacBio and Illumina data and correct small indels. Raw PacBio reads were mapped onto the completed genome with BLASR (*Chaisson & Tesler, 2012*). The corrected genome was re-annotated with Prokka and uploaded to the Integrated Microbial Genomes and Microbiomes (IMG/M) system of the Joint Genome Institute (JGI) for independent annotation (*Chen et al., 2019*).

Twenty-eight fragmented pairs of genes were subject to additional manual curation and correction where a pyrrolysine or selenocysteine residue had been erroneously translated as a stop codon (Table S3). The JGI annotation was publically updated to reflect these manual annotations, and this annotation was used for all genomic analyses. The genome has subsequently been re-annotated by NCBI. An overview of the genome assembly and annotation pipeline is provided in Fig. 1.

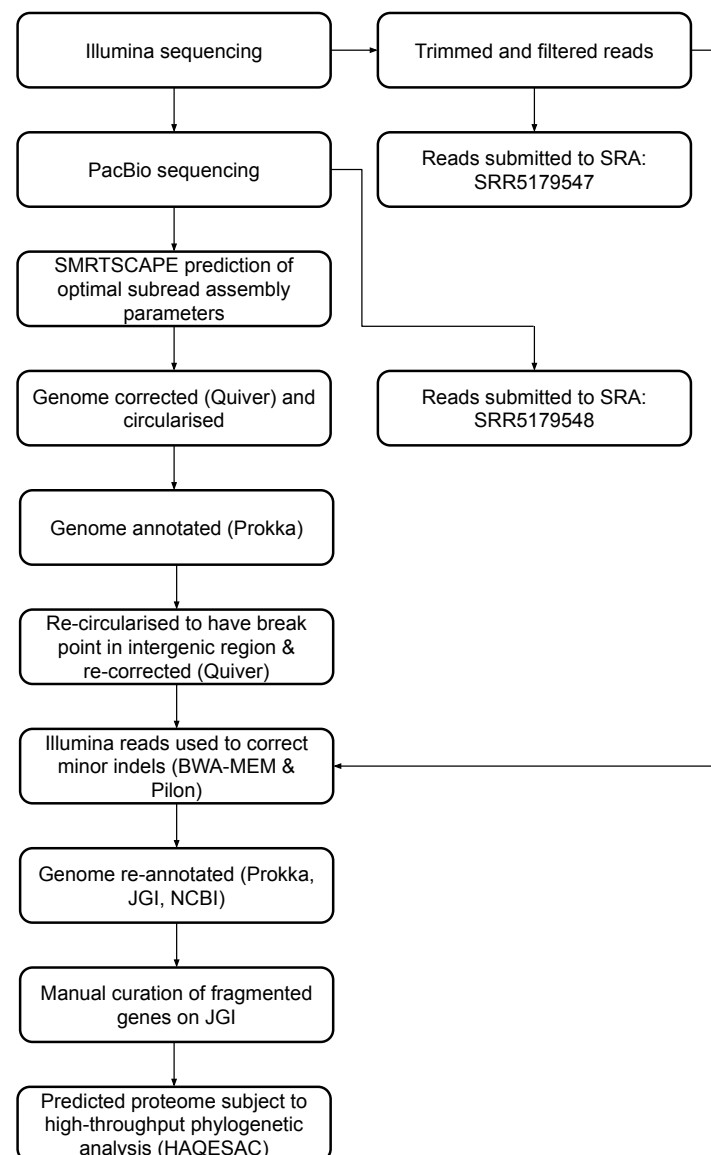

**Figure 1 DCMF genome assembly and annotation pipeline.**

## 16S rRNA gene identification and phylogeny

The DCMF 16S rRNA gene consensus sequence was searched against the NCBI prokaryotic 16S rRNA BLAST database as well as the 16S rRNA gene sequences of the two other known DCM-fermenting bacteria (absent from that database), *D. formicoaceticum* strain DMC (NCBI locus tags CEQ75_RS05455, CEQ75_RS05490, CEQ75_RS13675, CEQ75_RS13970, CEQ75_RS17045) and 'Ca. Dichloromethanomonas elyunquensis' strain RM (KU341776.1). The closest phylogenetic relatives and an outgroup, *Moorella perchloratireducens* strain An10 (NR_125518.1), were aligned with MAFFT program v.7 (*Kuraku et al., 2013*) and a neighbour-joining tree constructed with 1000 bootstraps

resampling a 200 PAM/k = 2 scoring matrix using 1,365 nucleotides. This was performed using Archaeopteryx (*Han & Zmasek, 2009*), as well as manual curation. In addition, DCMF 16S rRNA gene sequences were mapped to taxa using the SILVA Alignment, Classification and Tree (ACT) Service (*Pruesse, Peplies & Glöckner, 2012*) with default values.

In order to identify any non-DCMF 16S rRNA genes from the PacBio sequencing data, all contigs from all of the attempted assemblies were combined into a single file. DCMF contigs were identified and removed and the remaining contigs reduced to a set that were non-redundant at the level of 99% global sequence identity for the shorter contig, using GABLAM (*Davey, Shields & Edwards, 2006*). Where redundancy was identified, the longer contig was retained. In total, 20,201 contigs were reduced to 1,538 non-redundant non-DCMF contigs, hereon referred to as "NR contaminants". Cohabitant bacteria 16S rRNA gene sequences were identified from NR contaminants using barrnap v0.9 (implementing HMMer v3.2.1 and bedtools v2.27.1). Sequences were mapped to taxa using the SILVA Alignment, Classification and Tree (ACT) Service (*Pruesse, Peplies & Glöckner, 2012*) with default values.

## High throughput phylogenetic analysis of predicted proteome

JGI-annotated DCMF proteins were further annotated via high-throughput homology searching, multiple sequence alignment and molecular phylogenetics using HAQESAC v1.10.2 (*Edwards et al., 2007*). BLAST+ v2.6.0 blastp (*Camacho et al., 2009*) was used to search each protein against three protein datasets: (1) all bacterial proteins in the UniProt Knowledgebase (*The UniProt Consortium, 2017*) (downloaded 2017-02-06); (2) the predicted DCMF proteome; (3) the nine NCBI proteomes available for closely related bacteria identified from 16S rRNA gene analysis: *D. formicoaceticum* (GCF_002224645.1), *Desulfosporosinus acididurans* (GCF_001029285.1), *Desulfosporosinus acidiphilus* (GCF_000255115.2), *Desulfosporosinus orientis* (GCF_000235605.1), *Desulfosporosinus hippei* (GCF_900100785.1), *Desulfosporosinus lacus* (GCF_900129935.1), *Desulfitobacterium metallireducens* (GCF_000231405.2), *Desulfitobacterium hafniense* (GCF_000021925.1), *Dehalobacter restrictus* (GCF_000512895.1). The top 50 blastp results for each dataset were combined and up to 60 homologues meeting the HAQESAC default filtering criteria were aligned with Clustal Omega v1.2.2 (*Sievers & Higgins, 2017*). Neighbour-joining phylogenetic trees (1,000 bootstraps) were inferred using ClustalW v2.1 and midpoint-rooted using HAQESAC. Paralogous subfamilies arising from gene duplications were identified as nodes where the two ancestral clades each had at least two different species and shared at least one of those species. Multiple sequences from the same species within one of these paralogous subfamilies were identified as "in-paralogues" (lineage-specific duplications) or possible sequence variants. DCMF in-paralogues were kept. Possible in-paralogues or sequence variants from other species were restricted to the single closest homologue to the DCMF query. NCBI annotated proteins were subsequently subjected to the same pipeline with the addition of the JGI predicted proteome to the search database.

Putative taxonomic assignments for each JGI protein were made using an in-house tool, TaxaMap (http://rest.slimsuite.unsw.edu.au/taxamap). TaxaMap identifies the smallest

clade to which the query DCMF protein can be confidently assigned by stepping ancestrally through the tree until it reaches a branch with a bootstrap support of at least 50% and at least one non-DCMF protein. If the root is reached without meeting these requirements, the full HAQESAC tree was used. Once the clade has been identified, all Uniprot species codes for that clade are extracted as putative taxonomic assignments. These are mapped onto parent species, genus, family, order, class and phylum classifications using UniProt Knowledgebase taxonomy. At each taxonomic level, the taxa list is reduced to be non-redundant and each taxon contributes equally, to reduce sampling biases. Where a species code could only be mapped to a higher taxonomic level, it was designated as an unknown taxon associated with that higher level, e.g., "Firmicutes fam." would indicate an unknown family within the phylum Firmicutes. Where no non-DCMF homologues were found, a protein was assigned "None". TaxaMap Assignments were made for each protein individually and then combined using two strategies: (1) Unweighted; (2) Bootstrap weighted. The unweighted assignment simply adds up the number of proteins assigned to a particular taxon. Where a protein is assigned to multiple taxa, each is given an equal proportion of that protein, e.g., if a protein mapped ambiguously to five taxa, each would receive 0.2 for that protein. Any taxa with a combined score below 1.0 across all proteins was excluded, and scores recalculated iteratively. For the weighted score, counts were multiplied by the percentage bootstrap support for the clade, e.g., if a protein was assigned to two taxa and the bootstrap support for the clade was 80%, each taxon would receive a score of 0.4 ($= 0.5 \times 0.8$).

A subset of eight core house-keeping genes and 47 ribosomal proteins (Table S4) was subject to re-analysis using Maximum Likelihood trees (1,000 bootstraps) inferred by IQTree v1.6.1 using ModelFinder (*Nguyen et al., 2015*; *Kalyaanamoorthy et al., 2017*).

## Genomic analysis

CheckM (*Parks et al., 2015*) was used to assess the completeness and contamination in the DCMF genome. SPADE (*Mori et al., 2019*) was used to analyse repeat regions in the genomes, using default parameters.

The 81 full-length predicted trimethylamine (TMA) methyltransferase protein sequences were aligned with MAFFT v7.310 (*Katoh et al., 2002*) and a Maximum-Likelihood tree (1,000 bootstraps) inferred by IQTree v1.6.1 using ModelFinder (*Nguyen et al., 2015*; *Kalyaanamoorthy et al., 2017*). Global pairwise percentage identities were calculated using GABLAM v2.28.2 (*Davey, Shields & Edwards, 2006*) from an all-by-all BLAST 2.5.0+ blastp search (*Camacho et al., 2009*).

Putative selenocystine-containing proteins were verified via multiple lines of evidence. The presence of a selenocysteine insertion sequence (SECIS) was confirmed in either the JGI annotation, or via bSECISearch (*Zhang & Gladyshev, 2005*). Glycine/betaine/sarcosine reductase genes were checked for the presence of the conserved cysteine(s) present either before (CxxU in *grdA*) (*Kreimer & Andreesen, 1995*) or after (UxxCxxC in *grdBFH*) the selenocysteine residue (*Wagner et al., 1999*). In order to determine the substrate specificity of these reductases, the predicted proteins encoding the two subunits of B component (GrdB/F/H and GrdE/G/I) were aligned with known glycine, glycine betaine, and sarcosine reductase B components from *Clostridium sticklandii*, *Peptoclostridium acidaminophilum*,

*Peptoclostridium litorale*, *Sporomusa ovata* An4, and *Sporomusa ovata* H1 with MUSCLE in UGENE v.1.32 (*Okonechnikov et al., 2012*) and an unrooted Maximum Likelihood tree (1,000 bootstraps) was inferred by IQ-Tree v1.6.1 using ModelFinder (*Nguyen et al., 2015*; *Kalyaanamoorthy et al., 2017*).

# RESULTS

## Enrichment of DCMF in DFE

Five 1% transfers (T1–T5) of the previously reported enrichment culture DCMD (*Lee et al., 2012*) were carried out. The initial three transfers produced methane in a molar ratio of 0.6 moles per mole of DCM (Fig. 2A). Addition of BES to the culture medium in T4 caused methanogenesis to cease, and T5 could utilise DCM without the generation of methane in the absence of BES (Fig. 2B). The absence of methanogenic populations was confirmed via archaeal specific PCR. While a clear band at ~660 bp was observed in a positive control and T3 culture, there was no archaeal PCR product from the enrichment culture after the addition and subsequent removal of BES. The non-methanogenic, DCM-fermenting enrichment culture was henceforth called DFE.

T5 was then subject to two rounds of dilution to extinction. Community diversity was monitored throughout these transfers by DGGE, which showed a trend towards purity, culminating in the presence of a single band from the lowest active dilution series culture ($10^{-3}$; Fig. S1). Sequencing of the primary band had the highest identity match to an uncultured *Peptococcaceae*, henceforth referred to as "DCMF".

The shift away from the *Dehalobacter* population originally shown to be linked to DCM-degradation (*Lee et al., 2012*), was confirmed with qPCR. The *Dehalobacter* sp. 16S rRNA gene was below the limit of detection ($1.45 \times 10^3$ copies $ml^{-1}$) at all stages of growth in DFE cultures after the removal of methanogenic populations.

## Genome assembly and annotation

Attempts were initially made to sequence the dominant, DCM-degrading organism using Illumina short read technology, which yielded 5,040,903 filtered read pairs for a total of 1,827,383,271 bp. However, the presence of the additional organisms in the DFE culture and lack of a reference genome hindered this approach. Instead, a pure PacBio long read strategy was used to assemble a full-length gap-free circular genome for DCMF. Trimmed and filtered Illumina reads (average 242× coverage) were used for final, minor error correction. The final genome assembly had an average of 132× PacBio coverage (min >50×) and no regions of unusual read depth (Fig. 3A). The genome was circularised at overlapping ends and every base was covered by long reads spanning at least 5 kb 5′and 3′(Fig. 3B). In addition to these assessments, CheckM evaluated the genome as 97% complete with a contamination rate of 2%.

The DCMF genome is 6,441,270 bp long and has a G+C content of 46.44% (JGI genome ID 2718217647; GenBank accession CP017634.1). JGI annotation initially revealed 5,801 predicted protein-coding genes. Manual curation of the 28 pairs of genes fragmented by the presence of the amino acids pyrrolysine and selenocysteine (encoded by in-frame UAG

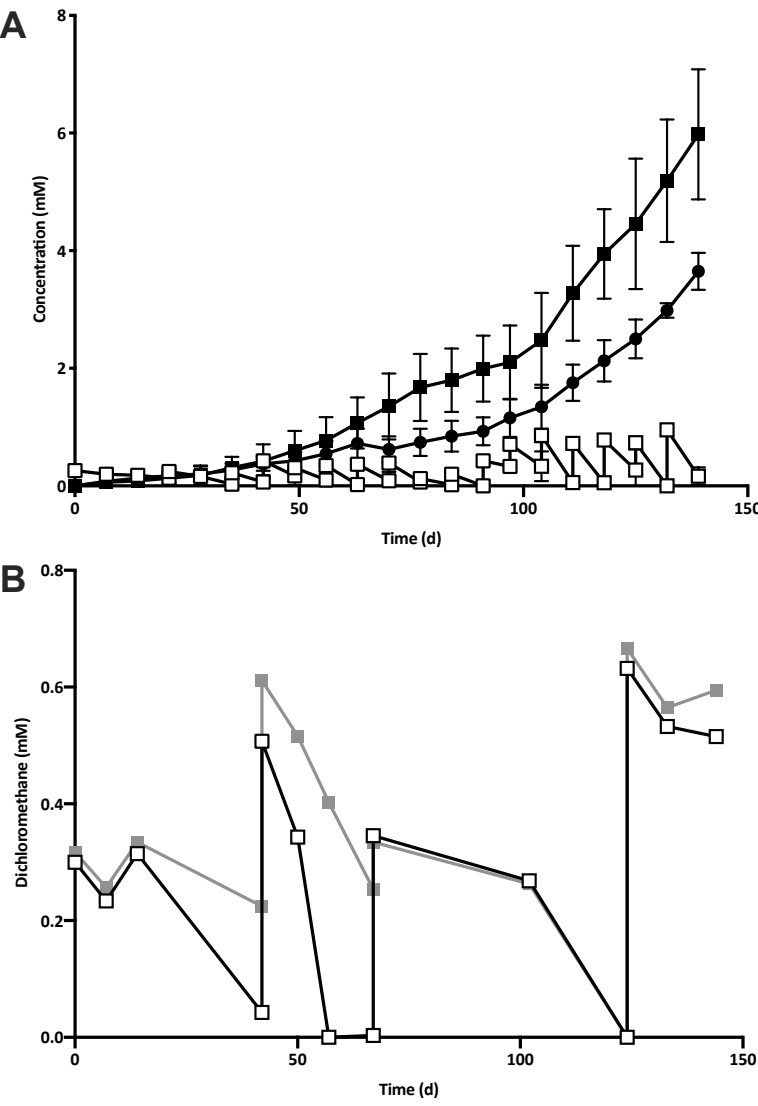

**Figure 2** **The removal of the methanogenic population from the DCM dechlorinating culture.** (A) The initial three transfers (T1–T3) of DCMD produced methane (black circles) in a molar ratio of 0.6 moles per mole DCM. DCM is shown both as actual concentration over time (white squares) as well as the cumulative DCM consumed (black squares). (B) DCM continued to be consumed in the presence (grey squares, subculture T4) and absence (white squares, subculture T5) of 2-bromoethanosulfonate, which caused methane production to cease.

and UGA stop codons, respectively; Table S3) brought this total down to 5,773 protein coding genes (Table 1).

## 16S rRNA gene phylogeny

The DCMF genome contains four full-length 16S rRNA genes (JGI locus tags Ga0180325_11664, 11677, 113771, 114507; Table S5), which share 99.87% identity when aligned. Based on the consensus 16S rRNA gene sequence, the closest relative to DCMF is *D. formicoaceticum* strain DMC (94% identity). This is closely followed by '*Ca.*

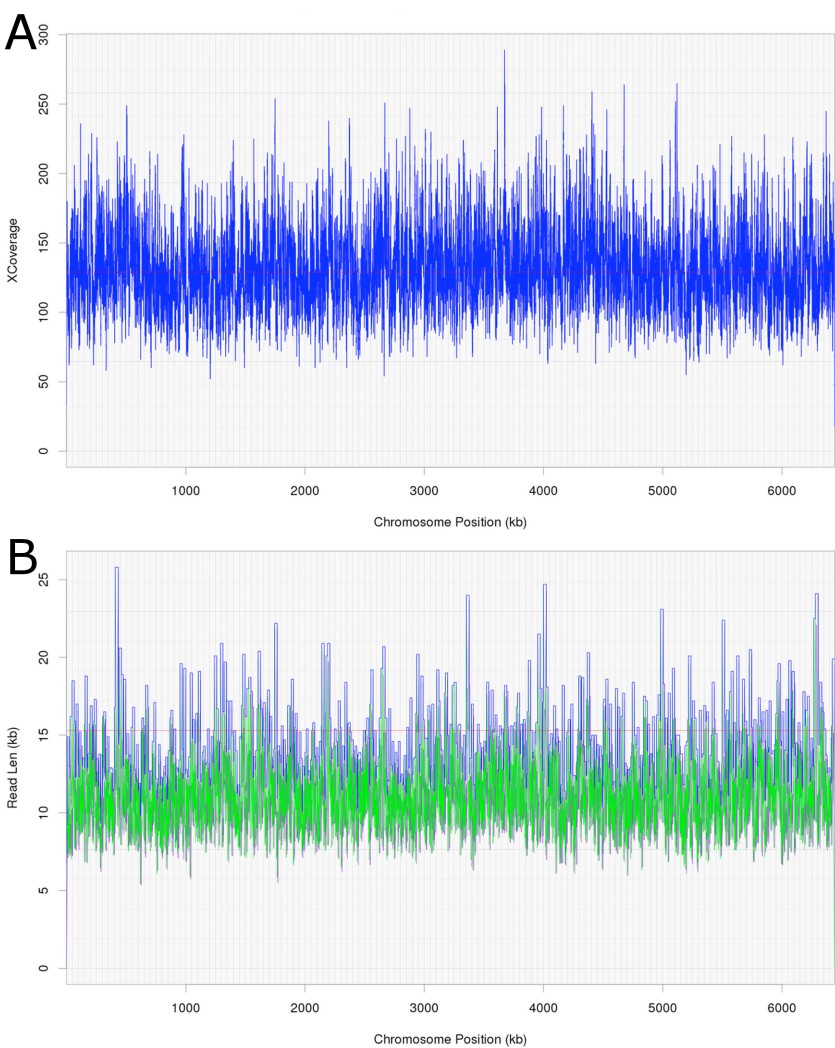

**Figure 3** **Average coverage depth and read length across the DCMF genome assembly.** (A) PacBio read depth along the full DCMF chromosome. Horizontal lines mark median depth (132×), and gradations as 1/8 median depth. (B) Maximum PacBio read length (kb) spanning each base along the full DCMF chromosome. Horizontal lines mark median length (15.3 kb), and gradations as 1/8 median length. Colours indicate total read length (blue), longest 5′ distance from base spanned by a single read (purple), and longest 3′ distance from base spanned by a single read (green).

Dichloromethanomonas elyunquensis' strain RM, *Dehalobacter restrictus* strain PER-K23 and *Desulfosporosinus acidiphilus* strain SJ4 (all 89% identity), and *Desulfitobacterium dehalogenans* strain ATCC 51507 (88% identity) (Fig. 4). The lowest taxonomic rank of SILVA classification was the family *Peptococcaceae*.

From the NR contaminants, 17 16S rRNA sequences were identified by barrnap. Based on SILVA classification, the 17 sequences were clustered into five distinct classifications (identified here by their lowest classified taxonomic rank): *Synergistaceae*, *Spirochaetaceae*, *Desulfovibrio*, *Ignavibacteria*, and *Lentimicrobiaceae* (Table S5). These were supported by clear clades within the SILVA tree (Fig. S2).

**Table 1    Comparison of the genomes of DCM-fermenting bacteria.**

| | "DCMF" | *Dehalobacterium formicoaceticum* | "*Candidatus* Dichloromethanomonas elyunquensis" |
|---|---|---|---|
| *Genome accession* | 2718217647 (JGI) | CP022121.1 (Genbank) | LNDB00000000.1 (Genbank) |
| *Genome size (bp)* | 6,441,270 | 3,766,545 | 2,076,422 |
| *G+C content (%)* | 46.4 | 43.2 | 43.5 |
| *Contigs* | 1 | 1 | 53 |
| *Protein-coding sequences* | 5,773 | 3,935 | 2,323 |
| *Metabolic pathways/genes of interest* | | | |
| Wood-Ljungdahl pathway | + | + | + |
| Reductive dehalogenases | − | − | + |
| Cobalamin biosynthesis | + | + | − |
| Glycine/betaine/sarcosine reductase complex | + | + | − |
| Methylamine methyltransferases | + | + | + |
| Pyrrolysine biosynthesis | + | + | − |
| *Reference* | This study | Chen et al. (2017) | Kleindienst et al. (2019) |

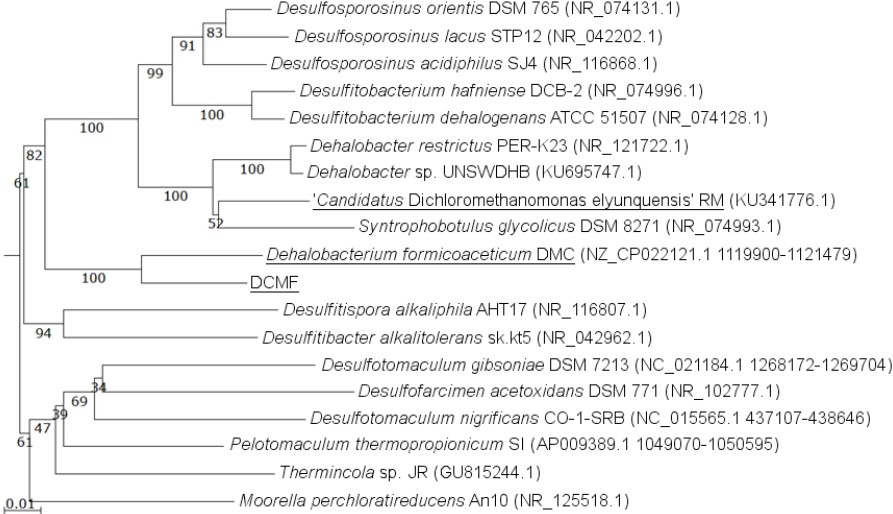

**Figure 4    16S rRNA gene phylogenetic tree of DCMF with closely related bacteria (94–87% identity).** Known DCM-fermenting bacteria are underlined. Numbers indicate percentage of branch support from 1,000 bootstraps. The scale bar indicates an evolutionary distance of 0.01 amino acid substitutions per site. Sequence were aligned in MAFFT program v.7 and a neighbor-joining tree (1,000 bootstraps) resampling a 200 PAM/k =2 scoring matrix was inferred using Archaeopteryx, with manual curation.

## Phylogenetic analysis of the predicted proteome

Taxonomic analysis of the whole predicted DCMF proteome was inconclusive at the genus level but strongly supported assignment within the order *Clostridiales* (Fig. 5). The top-ranked genus was *Dehalobacterium* (25.7% proteins, bootstrap-weighted), supporting the 16S rRNA gene phylogeny (Fig. 4) with *D. formicoaceticum* as the closest known relative of DCMF. The top families were *Peptococcaceae* (39.3%) and *Clostridiaceae* (11.2%).

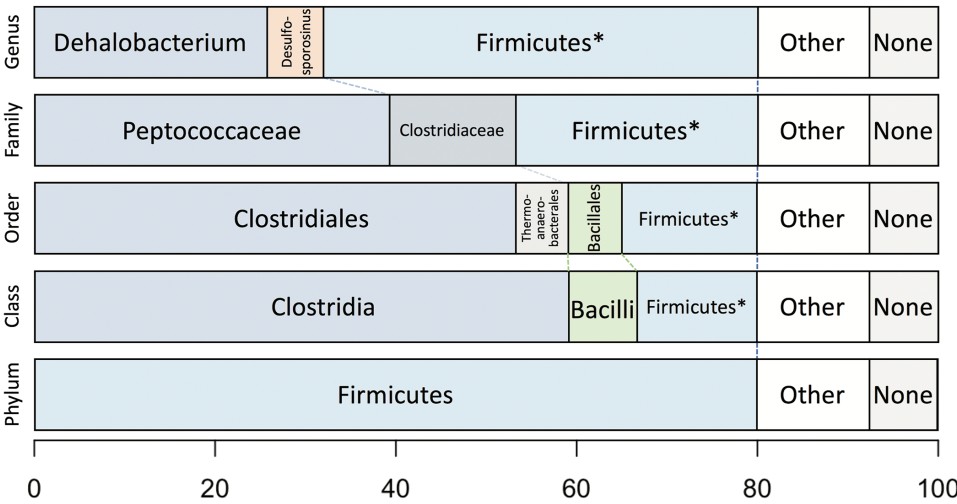

**Figure 5** **Bootstrap-weighted combined taxonomic assignments for the DCMF predicted proteome based on TaxaMap processing of high-throughput phylogenetic analysis.** Results are shown at five taxonomic levels: genus, family, order, class and phylum. The asterisk (*) indicates where low abundance and/or unknown Firmicutes taxa have been combined at the genus, family, order and class levels.

Whole-proteome TaxaMap analysis provides a good overview but is clearly influenced by the availability of homologous sequences in the search databases and may also be disrupted by, for example, horizontal gene transfer. We therefore restricted analysis to a more robust set of eight house-keeping genes and 47 ribosomal proteins (Table S4). With the exception of one malate dehydrogenase (Ga0180325_112460) and SSU ribosomal protein S10P (Ga0180325_114571), all proteins support *D. formicoaceticum* as the closest known relative of DCMF and placement in the *Peptococcaceae* family. All 55 genes support placement in *Clostridiales* (Table S4). Multiple sequence alignments, phylogenetic trees and TaxaMap assignments for all proteins can be found in online supplementary material at: http://www.slimsuite.unsw.edu.au/research/dcmf/. The restricted housekeeping genes can be found at: http://www.slimsuite.unsw.edu.au/research/dcmf/dcmf-hk.php.

## Genomic features of DCMF

A number of metabolic pathways were identified in the DCMF genome (Table 1, Fig. 6). The most prominent of these is the full set of genes for the Wood-Ljungdahl pathway (Fig. 6, Table S6). No reductive dehalogenases were identified in the genome by any of the three independent annotation pipelines.

The genome also contains an abundance of methylamine methyltransferase genes (Table S6), including 82 copies of TMA methyltransferase, *mttB*. There is a high diversity amongst the *mttB* genes, with an average amino acid sequence identity of only 30.3%. Associated with the presence of these methyltransferases are all five genes necessary to synthesise and utilise pyrrolysine (Table S8 ), a non-canonical amino acid residue present in 23 of the 96 total methylamine methyltransferases in the genome. In a maximum likelihood phylogenetic tree contructed from the 81 full-length *mttB* genes, the pyrrolysine-containing copies tend to cluster together at the bottom of the tree (Fig. S3). The pyrrolysine gene

Peer J

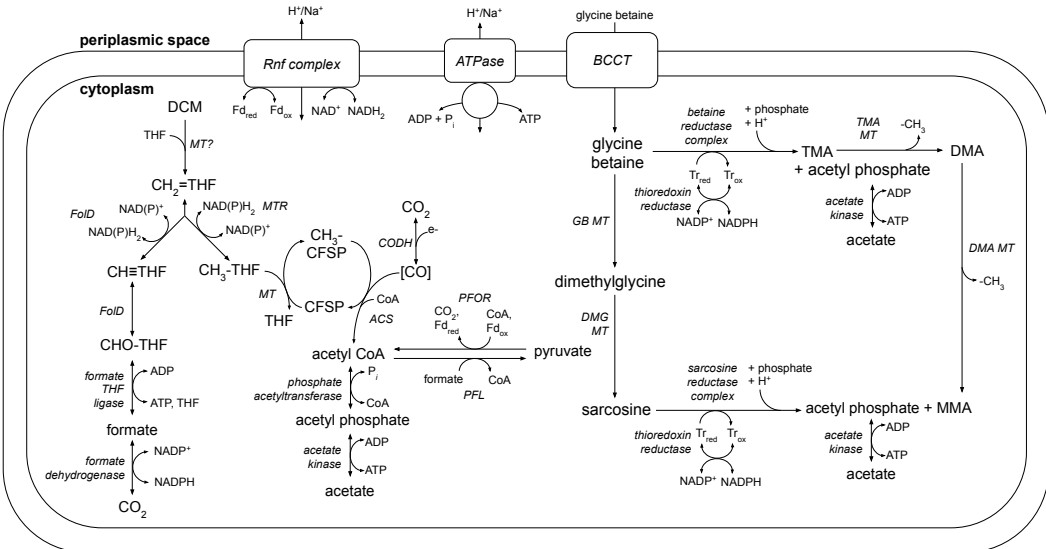

**Figure 6  A genome-based metabolic model for DCM and amine catabolism in DCMF.** The known substrate DCM is proposed to be transformed via the Wood-Ljungdahl pathway (left). The genome also suggests that DCMF may be able to utilize one or more of: tri-, di- and monomethylamine, glycine betaine, dimethylglycine, and sarcosine (right). Both sides of the model are predicted to produce acetyl compounds that can be transformed into pyruvate, linking them with the central carbon metabolism of the cell. Enzymes are written in italics. Gene loci for each enzyme are listed in Table S6. Abbreviations: ACS, acetyl CoA synthase; BCCT, betaine/choline/carnitine transporter; DCM, dichloromethane; DMA, dimethylamine; DMG, dimethylglycine; (CH$_3$)CFSP, (methyl) Co(I) corrinoid iron-sulfur protein; CH$\equiv$THF, 5,10-methenyl-tetrahydrofolate; CH$_2$ $=$ THF, 5,10-methylene-tetrahydrofolate; CH$_3$-THF, 5-methyl-tetrahydrofolate; CHO-THF, formyl-tetrahydrofolate; CODH, carbon monoxide dehydrogenase; Fd, ferredoxin; FolD, bifunctional 5,10-methylene-tetrahydrofolate dehydrogenase/5,10-methylene-tetrahydrofolate cyclohydrolase; MMA, monomethylamine; MT, methyltransferase; MTR, 5,10-methylene-tetrahydrofolate reductase; ox, oxidized; PFL, pyruvate formate lyase; PFOR, pyruvate-ferredoxin oxidoreductase; red, reduced; THF, tetrahydrofolate; TMA, trimethylamine; Tr, thioredoxin.

cluster in DCMF includes the dedicated tRNA (*pylT*), tRNA synthetase (*pylSc* and *pylSn*), and associated biosynthetic enzymes (*pylBCD*) (reviewed in (*Krzycki, 2013*)).

The presence of all genes required for *de novo* corrinoid biosynthesis (Table S9) is pertinent both to certain Wood-Ljungdahl pathway proteins and the methylamine methyltransferases, which require a corrinoid cofactor to function (*Burke & Krzycki, 1997*; *Ferguson et al., 2000*). However the genes for methionine synthesis (*metH* and *metE*), required to form S-adenosylmethionine, which is in turn used as a methyl donor during corrin ring formation (*Deeg et al., 1977*), were not identified in the genome. DCMF may be using an alternative route for *de novo* biosynthesis of this amino acid.

Additionally, five clusters of glycine/sarcosine/betaine reductase complex genes were found (Fig. 6, Table S6), indicating that DCMF may have a wider metabolic repertoire than its obligate DCM-fermenting relatives. Four of these clusters include the thioredoxin reductase (*trxB*) and thioredoxin I (*trxA*) necessary for electron transfer to the reductase, and the genome also encodes betaine/choline/carnitine transporter genes necessary to import these compounds into the cell. Components A (*grdA*) and B (*grdBE/FG/HI*) of the

glycine/sarcosine/betaine reductase complex contain an integral selenocysteine residue, as does the formate dehydrogenase (Ga0180325_112876, 112877, 112878s80) encoded in the DCMF genome. Accordingly, the organism contains the full complement of genes necessary for biosynthesis and incorporation of the unusual amino acid selenocysteine (*selABD, serS*; Table S7). All predicted selenocysteine-containing proteins also contain the SECIS downstream of the UGA stop codon, necessary for translating it as a selenocysteine residue instead.

## DISCUSSION

### The shift from a *Dehalobacter* species to DCMF

The novel *Peptococcaceae*, DCMF, was enriched from a previously reported methanogenic consortium, DCMD, where DCM was supplied as the sole energy source (*Lee et al., 2012*). That consortium was dominated by a *Dehalobacter* species whose growth was linked to DCM metabolism, producing acetate and methane. The Archaeal population was dominated by a hydrogenotrophic methanogen from the genus *Methanoculleus*. Furthermore, *Dehalobacter* sp. growth could be inhibited by addition of excess hydrogen. These two phenomena led to the conclusion that hydrogen was a DCM fermentation product along with acetate, and that a syntrophic association existed between *Dehalobacter* and *Methanoculleus* (*Lee et al., 2012*). In the present study, inhibition of methanogens with BES likely led to increased hydrogen levels, which inhibited the growth of the *Dehalobacter* sp. and enabled the hitherto unknown non-hydrogenogenic DCMF to become the dominant DCM fermenter in the DFE culture.

The cohabiting bacteria in the DFE culture have persisted despite attempts to isolate DCMF. These have been limited to serial transfers of dilution to extinction, due to the inability of the organism to form colonies on agar plates or in semi-solid agar shakes. Nonetheless, this has lead to a highly enriched culture, with community fingerprinting results showing only a single lineage.

### Optimisation for a high quality genome assembly from a mixed culture

Based on the 16S rRNA gene sequence retrieved from the DGGE community analysis, DCMF appeared to be an organism with comparatively few cultured relatives. Thus, whole genome sequencing was carried out in order to learn more about its role and function in the enrichment community. The lack of a reference genome and other organisms in the enrichment culture hindered attempts to assemble the genome from short read sequences only, making the long read capability of PacBio sequencing indispensible for this effort. Although long reads are prone to a higher proportion of sequencing errors than short reads, a series of checks were put in place to ensure that a high quality, uncontaminated genome assembly was obtained.

The use of SMRTSCAPE to predict the optimal HGAP settings allowed rapid comparison of various assembly parameters. By increasing the minimum correction coverage from $6\times$ to $10\times$, the total size of the assembly (including contaminant organism DNA) decreased from ~16 Mb to ~8.8 Mb, while the size of the DCMF genome remained relatively stable

around 6.4 Mb. Increasing the minimum correction coverage one step further to 11× resulted in a significant reduction of the DCMF genome to 1.9 Mb, indicating that much of the assembly was likely being lost to overzealous correction (Table S2).

The large size of the DCMF genome distinguishes it from the two other known DCM-fermenting bacteria, *D. formicocaceticum* and "*Ca.* Dichloromethanomonas elyunquensis" (Table 1). When assembling a genome *de novo* from a mixed culture, there is always the concern that stretches of other contaminating genomes will be mis-incorporated into the assembly. This likelihood was reduced by our assembly strategy of increasing stringency. The consistent sequencing coverage across the final genome (Fig. 3) strongly indicates that there was no such mis-assembly. The CheckM contaminant rate of 2% further confirms that the large DCMF genome is not over-inflated due to contamination. Analysis of repeated sequence motifs with SPADE showed that they comprise just 21,395 bp (0.03%) of the total DCMF genome, which also rules this out as a source of the large genome size. JGI annotation predicted 5,773 protein coding genes, giving a gene density of approximately 0.9 genes per kilobase, which is consistent with normal bacterial gene density (*Koonin & Wolf, 2008*).

## Genome annotation quality and availability of data

Despite the numerous error limiting and quality control steps taken in this study, it is almost certain that some errors will remain in both the genome sequence and genome annotation. We have therefore provided rich supplementary data to enable rapid, detailed analysis of potential genes and proteins of interest. The DCMF genome is available for browsing via a public Web Apollo (*Lee et al., 2013*) genome browser, accessed via the supplementary data site: http://www.slimsuite.unsw.edu.au/research/dcmf/. Results of three annotation pipelines (Prokka, JGI and NCBI) are available through the browser for direct comparison, along with mapped PacBio reads for assessing genomic sequence quality. A search tool has also been provided, enabling Exonerate (*Slater & Birney, 2005*) or BLAST+ (*Camacho et al., 2009*) searches of cDNA, peptides or genomic DNA against the DCMF genome, with hits linking directly to the corresponding region of the Web Apollo genome browser. Furthermore, multiple sequence alignments and phylogenetic trees have been provided for every JGI- and NCBI- annotated protein, enabling rapid assessment of protein descriptions and completeness. The genome and all annotation, including multiple sequence alignments and phylogenetic trees have been provided for every protein, are available to download from the Open Science Foundation (*Edwards et al., 2019*).

## Central carbon and energy metabolism in DCMF

The genome of DCMF suggests that it dechlorinates DCM via incorporation into the Wood-Ljungdahl pathway in a similar manner to that suggested for the two other DCM-fermenting bacteria, *D. formicoaceticum* and '*Ca.* Dichloromethanomonas elyunquensis' (*Mägli, Messmer & Leisinger, 1998*; *Kleindienst et al., 2019*). All genes for this pathway were identified within the genome, as well as those linking acetyl CoA to pyruvate and central carbon metabolism within the cell (Fig. 6, Table S6). DCM catabolism via the Wood-Ljungdahl pathway would result in the production of acetate, and possibly formate.

The DCMF genome encodes a cytoplasmic formate dehydrogenase (Ga0180325_112876, 112877, 112878s80) that could theoretically oxidise the formate to $CO_2$, however it remains unclear whether it is functional within the cell. *Mägli, Wendt & Leisinger (1996)* found that, despite the presence of formate dehydrogenase activity in cell extracts, *D. formicoaceticum* could not further metabolise the formate it produced, and instead accumulated it with acetate in a 2:1 molar ratio.

The DCMF genome also encodes an $F_1 F_O$-type ATPase (Fig. 6; Table S6), suggesting that it could employ a chemiosmotic mechanism for energy conservation alongside substrate-level phosphorylation of DCM. *D. formicoaceticum* and '*Ca.* Dichloromethanomonas elyunquensis' also encode this ATPase; and evidence for its activity has been detected in cell-free extracts of the former and via proteomics for the latter (*Mägli, Messmer & Leisinger, 1998*; *Chen et al., 2017*; *Kleindienst et al., 2019*). DCMF may generate a proton- or sodium-motive force for this ATPase via the Rnf complex, an ion-motive ferredoxin-NAD oxidoreductase encoded in its genome (Fig. 6; Table S6). The Rnf complex is of particular importance given the absence of any electron-bifurcating hydrogenases in the genome. Typically, it pumps ions out of the cell, catalyzing electron transfer from reduced ferredoxin to $NAD^+$ (*Biegel, Schmidt & Müller, 2009*; *Biegel & Muller, 2010*), while the ATPase uses the flow of ions back into the cytoplasm to convert ADP to ATP. However, these two transmembrane protein complexes can also act in reverse in order to balance the pool of reduced electron carriers within the cell (e.g., (*Lechtenfeld et al., 2018*)).

## An abundance of methyltransferases may indicate key role in DCM and wider metabolism

The protein responsible for the dechlorination of DCM remains elusive within DCM-fermenting bacteria. However, there is growing evidence that a novel methyltransferase is responsible for transforming DCM into 5,10-methylene-THF, which then enters the Wood-Ljungdahl pathway (*Mägli, Messmer & Leisinger, 1998*; *Chen et al., 2018*; *Kleindienst et al., 2019*). DCMF encodes an abundance of predicted methyltransferase proteins in its genome, hinting at a key role in metabolism. There are 96 genes annotated as a component of a methylamine methyltransferase system. They comprise the methyltransferase I (MTI, the *mt_B* genes), which transfers a methyl group onto a substrate specific corrinoid protein (the *mt_C* genes), from which the methyltransferase II (MTII, *mtbA*) transfers the methyl group to the final receiving compound (coenzyme M in methanogenic archaea or THF in acetogenic bacteria) (*Van der Meijden et al., 1983a*; *Van der Meijden et al., 1983b*; *Sauer & Thauer, 1997*; *Sauer, Harms & Thauer, 1997*)).

The majority (82) of the 96 assorted methylamine methyltransferase genes in DCMF are annotated as component B of a TMA methyltransferase (*mttB*). The remaining genes include di- and monomethylamine methyltransferases (*mtbB* and *mtmB*), the cognate corrinoid proteins for tri-, di- and monomethylamine (*mttC*, *mtbC*, *mtmC*), and the non-specific corrinoid protein methyltransferase (*mtbA*) (Table S6).

The high diversity amongst the TMA methyltransferases in DCMF (average pairwise amino acid sequence identity of 30.3%) may indicate that these genes have diversified to accommodate cobalamin cofactors with various upper and lower ligands (*Visser et al.,*
*2016*) and/or that they have more than one function within the cell, i.e., that they are in fact methyltransferases for a wider variety of substrates than merely methylamines. It has previously been shown that the chloromethane dehalogenase CmuAB is functionally similar to the monomethylamine methyltransferase MtaA (*Studer et al., 2001*). Moreover, four corrinoid-dependent methyltransferases were highly expressed in the proteome of DCM-fermenting '*Ca.* Dichloromethanomonas elyunquensis'(*Kleindienst et al., 2019*), further indicating that the array of corrinoid-binding methyltransferases in DCMF, along with its complete corrinoid biosynthetic pathway, may be crucial to the metabolism of DCM.

Interestingly, 23 of the 96 methylamine methyltransferase genes contain a pyrrolysine residue, identifiable as an in-frame UAG (amber) stop codon. While the *mttB* gene is widespread amongst bacteria and archaea, most organisms do not encode the pyrrolysine residue (*Srinivasan, 2002*; *Ticak et al., 2014*). Indeed, the *pylTSBCD* gene cluster to synthesise and incorporate this non-canonical amino acid is limited to a small number of bacterial genera, including *Desulfotomaculum, Desulfitobacterium*, and *Thermincola* (*Gaston, Jiang & Krzycki, 2011*)—all members of the *Peptococcaceae* family and close relatives of DCMF based on 16S rRNA phylogeny. *D. formicoaceticum* also encodes the *pyl* genes, but ''*Ca.* Dichloromethanomonas elyunquensis'' does not (Table 1).

It is worth noting that '*Ca.* Dichloromethanomonas elyunquensis' is unique among the three DCM-fermenting species in encoding reductive dehalogenase genes in its genome (Table 1). This finding, coupled with a recent dual carbon-chlorine isotopic analysis of the two previously-reported DCM-fermenters (*Chen et al., 2018*), suggests that there are distinct DCM dechlorination mechanisms operating in these organisms. Based on the presence or absence of key pathways in the genome (Table 1) and phylogenetic analysis (Fig. 4), DCMF appears to have more in common with *D. formicoaceticum* than '*Ca.* Dichloromethanomonas elyunquensis'.

## A genome-based model for amine metabolism in DCMF

DCMF, *D. formicoaceticum*, and '*Ca.* Dichloromethanomonas elyunquensis' have thus far only been cultured on DCM (*Mägli, Wendt & Leisinger, 1996*; *Kleindienst et al., 2017*), but the larger genome of DCMF suggests that perhaps it has a wider substrate repertoire. As well as a potential role in DCM dechlorination, the many TMA methyltransferase genes could in fact allow the organism to use TMA as a substrate for growth. In a pathway akin to that outlined for *Sporomusa ovata* An4 (*Visser et al., 2016*), TMA could be demethylated in a stepwise manner to dimethylamine and monomethylamine (Fig. 6). Each methyl group released would be capable of providing six reducing equivalents when oxidised to $CO_2$.

It has also been shown that a nonpyrrolysine *mttB* homolog within *Desulfitobacterium hafniense* Y51 is in fact a glycine betaine methyltransferase, *mtgB* (*Ticak et al., 2014*). Glycine betaine methyltransferase systems (including the MTI, MTII, and cognate corrinoid proteins outlined above) have also been found in *S. ovata* An4 (*Visser et al., 2016*), *S. ovata* H1 DSM 2662, and *Acetobacterium woodii* (*Lechtenfeld et al., 2018*). In *S. ovata* An4, the same genes were proposed to further demethylate the resulting dimethylglycine to sarcosine (*Visser et al., 2016*).

When the glycine betaine MTI and MTII proteins from each of these organisms were searched against the DCMF predicted proteome, a number of homologs were identified that may be used for glycine betaine or dimethylglycine demethylation (Fig. 6). In the MttB maximum likelihood tree (Fig. S3), the top five highest identity homologs to the MtgB proteins from these species all clustered together in a distinct clade. Ga0180325_115483, annotated as a trimethylamine methyltransferase, had the highest percentage amino acid identity (54–55%) to the MTI proteins from all five species. However, it sits isolated on the opposite strand of DNA to all surrounding genes, thus making it an unlikely candidate. The second highest identity MTI homolog to all bar one species was Ga0180325_114740 (54–55% identity), which sits in a neighbourhood with three methyl-THF methyltransferases (shown to act as an MTII in *S. ovata* An4 (*Visser et al., 2016*)), two further TMA MTIs, a cognate corrinoid protein, and two betaine/choline/carnitine transporter genes. This genetic neighbourhood contains all the requisite components for the demethylation system, and is thus a good candidate for glycine betaine metabolism in DCMF.

The glycine betaine MTII proteins of the *Sporomusa* and *Acetobacterium* species had a lower percentage identity to the nearest homolog in DCMF (44–45% to Ga018325_111809), and even lower again to the second best hit (32–36% to Ga018325_111232). This makes it difficult to determine any definitive candidates for a glycine betaine or dimethylglycine MTII gene in DCMF. Nonetheless, relevant MTIs and methyltransferase cognate corrinoid proteins once again surround these lower identity homologs in the DCMF genome, making them further possible candidates for glycine betaine and/or dimethylglycine demethylation in DCMF.

In addition to demethylation, glycine betaine and sarcosine can be reductively cleaved to form tri- or monomethylamine, respectively, plus acetyl phosphate. The DCMF genome contains five separate clusters with glycine/betaine/sarcosine reductases and the thioredoxin I and reductase necessary for electron transfer to the reductase (Fig. 6, Table S6). These reductase genes are also present in *D. formicoaceticum*, but absent from "*Ca.* Dichloromethanomonas elyunquensis" (Table 1). The glycine/sarcosine/betaine reductase complex consists of three components: the selenocysteine-containing component A (GrdA); a two- or three-subunit component B, one of which contains a selenocysteine residue (GrdBE/FG/HI); and component C, which is post-translationally combined into a single protein (GrdCD) (*Andreesen, 2004*). In order to determine the substrate specificity of the component B genes in DCMF, they were aligned with known component B genes from *Clostridium sticklandii*, *Peptoclostridium acidaminophilum*, *Peptoclostridium litorale*, *S. ovata* An4, and *S. ovata* H1. Of the five B components in DCMF, Ga0180325_114802 (GrdG) and Ga0180325_114803s (GrdF) are likely specific to sarcosine, as they clustered with the sarcosine reductase genes from the two *Sporomusa* species, while Ga0180325_11115251 (GrdI) and Ga0180325_115252s54 (GrdH) clustered with the glycine betaine reductases (Fig. S4). One of the remaining B components is likely a pseudogene (Ga0180325_11855s), as it lacks the required UxxCxxC motif after the selenocysteine residue to protect against accidental oxidation (*Parther, 2003*). The substrate-specificity of the remaining two B components (Ga0180325_114453 and Ga0180325_114454s56; Ga0180325_114684 and Ga0180325_114685s86) is unclear, as they did not cluster with

any of the annotated reductase genes (Fig. S4). The DCMF genome therefore suggests that this organism may be able to metabolise methylated amine compounds via two different mechanisms, which could act in concert: reductive cleavage and demethylation (Fig. 6).

## CONCLUSIONS

DCMF is an organism that demonstrates a relatively rare metabolism and harbours a large genome. Both long and short read genome sequencing technology were used to compliment each other and assemble a singular, circular chromosome for the organism, despite the low-level presence of other bacteria in the enrichment culture. DCMF is the dominant organism in the enrichment and likely sits within the *Peptococcaceae* family, although not within any known genus. Its DCM-fermenting capabilities make it of interest to the bioremediation sector and the genome contains clues to the as-yet undiscovered DCM dechlorinating enzyme, the identification of which will be the subject of future work. The genome also suggests that DCMF can metabolise a number of amine compounds, such as methylated amines, glycine betaine, dimethylglycine, and sarcosine, though these capabilities remain to be tested. Extensive supplementary data for the DCMF genome and annotation is available at http://www.slimsuite.unsw.edu.au/research/dcmf/ and the Open Science Foundation (*Edwards et al., 2019*).

## ACKNOWLEDGEMENTS

We thank Dr Bat-Erdene Jugder (University of New South Wales) for his assistance with the DNA extractions for PacBio sequencing and Dr Xabier Vázquez-Campos (University of New South Wales) for assistance with data retrieval.

### Funding

Sophie I. Holland was supported by an Australian Government Research Training Program scholarship. The funders had no role in study design, data collection and analysis, decision to publish, or preparation of the manuscript.

### Grant Disclosures

The following grant information was disclosed by the authors:
Australian Government Research Training Program scholarship.

### Competing Interests

The authors declare there are no competing interests.

### Author Contributions

- Sophie I. Holland and Haluk Ertan performed the experiments, analyzed the data, prepared figures and/or tables, authored or reviewed drafts of the paper, approved the final draft.

- Richard J. Edwards conceived and designed the experiments, performed the experiments, analyzed the data, contributed reagents/materials/analysis tools, prepared figures and/or tables, authored or reviewed drafts of the paper, approved the final draft.
- Yie Kuan Wong conceived and designed the experiments, performed the experiments, analyzed the data, prepared figures and/or tables, approved the final draft.
- Tonia L. Russell and Nandan P. Deshpande performed the experiments, analyzed the data, contributed reagents/materials/analysis tools, approved the final draft.
- Michael J. Manefield and Matthew Lee conceived and designed the experiments, contributed reagents/materials/analysis tools, authored or reviewed drafts of the paper, approved the final draft.

### DNA Deposition

The following information was supplied regarding the deposition of DNA sequences:

The DCMF genome is available at JGI genome (which has both a log in and a public website option) ID 2718217647 (final annotation) and at GenBank: CP017634.1.

### Data Availability

The raw PacBio and Illumina sequencing reads are available at the NCBI Sequence Read Archive: SRR5179548 and SRR5179547.

The DCMF genome, results of multiple annotation pipelines, and mapped PacBio reads are available on a public Web Apollo genome browser, via the supplementary data site: Available at http://www.slimsuite.unsw.edu.au/research/dcmf/.

This data is also available at: Edwards, Richard J, Sophie I Holland, Haluk Ertan, Michael Manefield, and Matthew Lee. 2019. "Holland et Al (2019) Supplementary Data." OSF. July 25. DOI:10.17605/OSF.IO/BK5MU.

The raw data for Fig. 1 is available in Table S10.

### Supplemental Information

Supplemental information for this article can be found online at http://dx.doi.org/10.7717/peerj.7775#supplemental-information.

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
