# Peer review of "Whole genome sequencing of a novel, dichloromethane-fermenting Peptococcaceae from an enrichment culture"

_PeerJ, doi:10.7717/peerj.7775_

## Round 0.1 · original submission · Major Revisions

There is a number of important issues raised by the reviewers regarding both the exposition and the analysis. They need to be addressed. One point that seems strange, is predicted pyrrolysines: it has been my understanding that they are limited to a certain group of archaea, and if they are found elsewhere, it in itself is an interesting fact. If pyrrolysines and selenocysteines indeed are predicted to occur in the studied species, this needs to be supported by identification of functionally linked genes (dedicated tRNAs, aminoacyl-tRNA synthetases, necessary biosynthetic enzymes).

Reviewer 1 ·

Basic reporting

Introduction seems very laconic to me. I suggest that you add a little bit more information about DCM degradation (line 46), and briefly describe Wood-Ljungdahl pathway or at least add the reference to the article where it is described in detail (line 52). I also suggest you to more heavily emphasize an importance of studying DCM-fermenting bacteria and its ecological impact.

Experimental design

Methods seem to be adequate and are described very thoroughly, with sufficient details to replicate. Though I think that some kind of basic visualization of the research pipeline would be useful and would make it easier to follow.

Validity of the findings

No comment

Additional comments

Statement about methylamine methyltransferases, which need a corrinoid cofactor to function, requires reference (line 310-311).
I would also suggest you to give more detailed focus to the metabolic abilities of the DCMF, explain how glycine/sarcosine/betaine reductases (line 301, 384) are connected to TMA utilization, how exactly methionine connects to the corrin ring formation (line 311-312) etc.

·

Basic reporting

The article is written in unambiguous, professional English. Although the information provided in introduction is not sufficient to understand the study. What is more, some of the information seems to be incorrect. Not all the necessary literature is mentioned in the introduction.
1. Lines 40-42: “Approximately 70% of 41 all DCM worldwide is of anthropogenic origin”. To confirm this fact author cite Marshall & Pottenger, 2016. There is no such statement in the mentioned article.
2. Lines 46-47: “To date, only two DCM-48 fermenting bacteria have been described” This claim seems suspicious, at least at Muller et al., 2011 (Research in Microbiology) at lot of such bacteria is mentioned.
In the same review (Muller et al., 2011) proposed mechanisms of DcmA action is described. It is improtant to mention in introduction that some details about DCM degradation are known.
3. Although in the introduction authors mention that methyl group have to be incorporated into Wood-Ljungdahl pathway, but do not mention that the presence of this pathway is not sufficient for DCM-degradation.
4. Also the important information for understanding the background is provided in the first paragraph of discussion section (lines 317-326). I suggest to move this to introduction.
5. The links to the raw Illumina and PacBio reads are provided. Reads are deposited in the SRA. Also authors provide link to the annotation files, that are surprisingly stored on their local server. In the Table 1 the accession to GenBank is also provided. But it seems it is annotated only with NCBI pipeline and does not include authors manual corrections. The GenBank Accession is crucial for genome availability for the scientific community, and though I think it will be important to include Accession in the main text and also possibly to update Genbank annotation with authors’ corrections.
6. The results provided are in general unclear. The part about methylamine methyltransferases seems unclear and unfinished.

Experimental design

The research is within Aims and Scope of the journal. The research question is not well defined, because only the fact that new bacteria is sequenced is provided, but no detailed metabolic reconstrction for the species is performed. I really appreciate here the part about genome assembly. The genome is accurately assembled from filtered reads, and the quality of assembly is accurately checked. Although some questions arise about manual gene corrections:
1. Why authors are so sure that the genes they corrected are indeed have pyrrolysines and selenocysteines, and not just pseudegenes? Especially, that most corrections were done in multi-copy genes.
2. As well it is not clear where the final annotation is stored, because on the website three independent annotations are provided.
3. The part about culture saturation is unclear. It is definitely part of previous experiment of archaea purification from the same region. The details for this part is hardly understandable for the general public. PeerJ considers articles in biological sciences in general, and that is why the process of obtaining the bacteria of interest have to be more clearly described, so it can be understood not only by specialists in the narrow field.
4. The selected method for taxonomy assignment is ok, but one expect more strait forward approach, such as comparison of 16S rRNA with SILVA database or RDP. It is unclear from methods what type of tree was build for 16S rRNA genes, and with what parameters.
5. The part about taxonomy assignment for individual proteins is unclear. I think the check for some some single-copy universal genes will be enough. Although those trees have to be ML ones with bootstraps.
6. In case of TMA methyltransferases it is unclear why not just to build multiple alignment and then phylogenetic tree. It will be easier to analyze clustered branches on such tree than from matrix provided by authors.
7. Why genomic DNA was extracted with the protocol developed for archaea DNA puification (line 97-98)?

Validity of the findings

Line 236: Abbreviation BES is not explained in the text.
Line 248: The idea about “shift away from the Dehalobacter population” is unclear. It have to be described in more details.
Line 271: it is unclear how contaminant bacteria were identified.
My main concern about this paper is the lack of metabolic reconstruction. The absence of reductive dehalogenases is mentioned in Table 1. Still it is unclear where from the list of potential dehalogenases was obtained?
Authors discuss that pyrrolysine biosynthesis pathway is present, but there is nothing about selenocysteine, although in the methods it was stated that corrections were made for both aminoacids.
Why methylamine methyltransferases are so important? Some of them can be associated with pyrrolysine biosynthesis, but what about others?
With what genes are those methyltransferases are surrounded? Such analysis may add some clues to the pathways behind DCM-utilization.
The more vivid metabolic reconstruction is important here because one of the main article claims is that the genome of the extracted bacteria have twice more genes than close relatives.
The discussion section is unstructured, although the description about genome assembly optimization is interesting.

Additional comments

The organism that you extracted seems to be interesting, but the article does not contain enough information to understand the importance of the finding.
First, the introduction does not provide enough information to understand the details and pathways behind DCM degradation. As well it will be useful to provide more explanations about Wood-Ljungdahl pathway.
Second, the sequenced genome itself is not that interesting. The more detailed analysis of this Peptococcaceae bacteria will add a lot to the text.
Third, some information have to be added in the text and not only mentioned in table 1, such as Genbank Accession and search for reductive dehalogenases.
Fourth, some methods used are not well-described. It is especially crucial for reproducing phylogeny reconstruction.
And last, there is no clear idea behind this text. The big part of manuscript is dedicated to optimization of the genome assembly, while less is discussed about sequenced bacteria properties. I think that the balance have to be shifted towards bacteria. That will add consistency to the text, and will be in better agreement with the abstract section.

---

## Round 0.2 · Minor Revisions

Both reviewers are satisfied with the revised manuscript. There remain only a couple of editorial glitches that may be easily corrected.

Reviewer 1 ·

Basic reporting

The article is well written, unambiguous professional English is used, all necessary references, figures and tables are provided.
Authors have taken previous comments into consideration and added some more information about DCM metabolism and Wood-Ljungdahl pathway into the Introduction section. Though it still remains a little laconic and not easy to completely understand without reading cited articles. I would advise to introduce the metabolic sheme given in Figure 6 much earlier, not in the middle of Discussion section. Thus it will show the connections between DCM, TMA, glycine/betaine/sarcosine etc right from the beginning and will make the whole picture more clear and structured.

Experimental design

Methods are adequate and are described thoroughly, with sufficient details to replicate. Research question is quite important from the ecological point of view.

Validity of the findings

All underlying data is provided, conclusions are well stated.

·

Basic reporting

I am satisfied with the changes authors made in this revision round.
I think that now this text has to be understandable by the general public.
Actually, the more detailed Discussion section now allows to better understand the unique properties of the isolated Bacteria.

Experimental design

The design is ok.
Now the methods are described in more details, so it can be reproduced.

Validity of the findings

The isolated DSMF bacteria is an interesting organism to study. Now when the obtained results are described and discussed in more details, I think that it will be evident to the general public, that this organism require more detailed biochemical analysis.
All the data is provided.

Additional comments

I really enjoyed reading the revised version of this manuscript, because now the findings are beautifully put into the wider scope and all the missing links are now explained.
I have only one small remark about Figure S2.
It seems like the font in this figure is somehow broken. And there is no explanation in this figure' legend about the black dots.

---

## Round 0.3 · accepted · Accept

All reviewers' concerns have been addressed.